# First Report of Single Nucleotide Polymorphisms (SNPs) of the Leporine Shadow of Prion Protein Gene (*SPRN*) and Absence of Nonsynonymous SNPs in the Open Reading Frame (ORF) in Rabbits

**DOI:** 10.3390/ani14121807

**Published:** 2024-06-17

**Authors:** Sameeullah Memon, Zerui Wang, Wen-Quan Zou, Yong-Chan Kim, Byung-Hoon Jeong

**Affiliations:** 1Korea Zoonosis Research Institute, Jeonbuk National University, Iksan 54531, Republic of Korea; sam.memon337@gmail.com; 2Department of Bioactive Material Sciences and Institute for Molecular Biology and Genetics, Jeonbuk National University, Jeonju 54896, Republic of Korea; 3Department of Pathology, Case Western Reserve University School of Medicine, Cleveland, OH 44106, USA; zxw488@case.edu (Z.W.); wenquanzou@ncu.edu.cn (W.-Q.Z.); 4Department of Biological Sciences, Andong National University, Andong 36729, Republic of Korea

**Keywords:** rabbit, leporine, *SPRN*, SNP, polymorphism, prion protein

## Abstract

**Simple Summary:**

The study focuses on the shadow of prion protein (Sho) encoded by the shadow of prion protein gene (*SPRN*), which associates with PrP and promotes the progression of prion diseases. While genetic polymorphisms in *SPRN* are linked to susceptibility in several species, rabbit SPRN gene polymorphisms have not been extensively studied. Through amplicon sequencing, we identified novel single nucleotide polymorphisms (SNPs) in the rabbit *SPRN* gene and found strong linkage disequilibrium (LD) between them. However, strong LD was not observed between polymorphisms in the prion protein gene (*PRNP*) and *SPRN* genes in rabbits. Comparison of amino acid sequences revealed differences in *SPRN* between rabbits and other species susceptible or resistant to prion diseases. This study represents the first examination of genetic features of the rabbit *SPRN* gene.

**Abstract:**

Prion disorders are fatal infectious diseases that are caused by a buildup of pathogenic prion protein (PrP^Sc^) in susceptible mammals. According to new findings, the shadow of prion protein (Sho) encoded by the shadow of prion protein gene (*SPRN*) is associated with prion protein (PrP), promoting the progression of prion diseases. Although genetic polymorphisms in *SPRN* are associated with susceptibility to several prion diseases, genetic polymorphisms in the rabbit *SPRN* gene have not been investigated in depth. We discovered two novel single nucleotide polymorphisms (SNPs) in the leporine *SPRN* gene on chromosome 18 and found strong linkage disequilibrium (LD) between them. Additionally, strong LD was not found between the polymorphisms of *PRNP* and *SPRN* genes in rabbits. Furthermore, nonsynonymous SNPs that alter the amino acid sequences within the open reading frame (ORF) of *SPRN* have been observed in prion disease-susceptible animals, but this is the first report in rabbits. As far as we are aware, this study represents the first examination of the genetic features of the rabbit *SPRN* gene.

## 1. Introduction

The prion protein (PrP) provides a vital contribution to the emergence of prion diseases, such as Creutzfeldt–Jakob disease (CJD), chronic wasting disease (CWD), scrapie, and bovine spongiform encephalopathy (BSE) [1]. The central nervous system in these disorders is characterized by the buildup of abnormal and misfolded prion proteins (PrP^Sc^), which causes neuron dysfunction and eventual death [2]. The prion protein gene *(PRNP)* is linked to the development of prion disorders through several genetic attributes. Recent studies have demonstrated that a number of cofactors associated with PrP promote the transformation of normal prion protein (PrP^C^) to PrP^Sc^ [3,4,5]. The shadow of prion protein (Sho), which is translated by the shadow of the prion protein gene (*SPRN*), is one of these cofactors and plays an important role in the progression of prion diseases [6]. The Sho is classified as a prion family protein and exhibits similarities to PrP because of a glycosylphosphatidylinositol (GPI) anchor and the presence of repeat domains [7]. As the Sho protein is a crucial collaborator in interactions with PrP, genetic variations in the *SPRN* gene that influence structural alterations of the Sho protein, and its expression physiology have been associated with susceptibility to prion disease.

The rabbit is one of the few species of mammals that appears to be immune to transmissible spongiform encephalopathy (TSE) agents [8]. To date, rabbits inoculated with scrapie, kuru, or CJD agents from mice or sheep have not shown symptoms of TSE diseases [9]. In scrapie-infected mouse neuroblastoma cells, which accumulate mouse PrP^Sc^, neither rabbit PrP^C^ nor chimeric rabbit-mouse PrP^C^ constructs have been transformed into the proteinase-K resistant form [1]. These results imply that the structural features of the leporine PrP are responsible for the resistance to conformational conversion to PrP^Sc^ and provide protection against TSE infection. The three-dimensional structure of leporine PrP can help us understand the unique characteristics that set it apart from PrPs of different species [8]. Sequence alignment also reveals 22 different amino acid residues between rabbit and mouse PrPs. The substitution of leporine-specific amino acids (N99G, L108M, N173S, or V214I) in mouse PrP inhibits abnormal conformational changes [10].

The *SPRN* gene was first identified in the 1990s when scientists began to speculate whether prion-induced disorders might be caused by an unidentified gene [7]. These scientists discovered a novel gene called “shadoo” or *SPRN*, which encodes a protein resembling the N-terminal region of the PrP. The amino acid sequences of Sho and PrP share a highly similar region with a hydrophobic alanine-rich sequence [5]. This PrP sequence plays a crucial role in interactions between PrP^C^ and PrP^Sc^ [11]. In the mammalian brain, the expression profiles of *SPRN* and *PRNP* overlapped, and a previous study indicated that the activity of the two genes was co-regulated [5]. Furthermore, in the brains of mice injected with a mice-adapted scrapie strain, there was a significant reduction in the endogenous expression of the Sho protein, suggesting a potential role of Sho in TSE development [12]. Patients with variant CJD carry a null allele associated with the *SPRN* gene [3]. Additionally, susceptibility to goat scrapie is linked to an insertion variant in the 3′ untranslated region (UTR) of the *SPRN* gene [4]. Similarly, L-type atypical BSE-affected cattle showed an infrequent sequence variation within the coding sequence of the bovine *SPRN* gene [5,6] These studies demonstrate a strong relationship between genetic polymorphisms of the *SPRN* genes and the pathological mechanisms of prion diseases. To date, polymorphisms of the *SPRN* gene have been examined in prion disease-sensitive and -resistant species [1,6,13,14,15,16,17]. However, the genetic polymorphisms of the leporine *SPRN* gene have not been explored in depth.

In the current investigation, we used amplicon sequencing and genotyping with *SPRN* gene-specific primers to examine genetic polymorphisms of the *SPRN* gene in 207 rabbits and examined the variation characteristics of the rabbit *SPRN* gene. We also assessed polymorphisms in the *SPRN* gene for linkage disequilibrium (LD) and investigated LD between *SPRN* and *PRNP* single nucleotide polymorphisms (SNPs). Additionally, we identified specific amino acids unique to rabbits by performing multiple sequence alignments with amino acid sequences of Sho protein in the eight prion-related species. Furthermore, we examined the frequencies of genetic polymorphisms within the open reading frame (ORF) region of the *SPRN* gene between rabbits and species susceptible to prion disease (humans, cattle, goats, and sheep) and those resistant to prion disease (horses and dogs).

## 2. Materials and Methods

### 2.1. Ethics Statements

The Institutional Animal Care and Use Committee at Jeonbuk National University (CBNU 2019-058) granted approval for all experimental methods. The Korean Experimental Animal Protection Act was followed for all experiments involving rabbits.

### 2.2. Genomic DNA Extraction

Blood samples were collected from 207 crossbred rabbits (New Zealand White and Flemish Giant rabbits) in the Nonghyup slaughterhouse in the Republic of Korea. The sample size of the study was sufficient to detect uncommon polymorphisms, including those with genotype frequencies below 1% [18]. Before analysis, ethylenediaminetetraacetic acid (EDTA)-treated whole blood was stored at −80 °C. Genomic DNA was extracted from 200 μL of whole blood using a QIAamp DNA Blood Mini Kit (Qiagen, Valencia, CA, USA) according to the manufacturer’s guidelines.

### 2.3. Polymerase Chain Reaction (PCR) and DNA Sequencing

PCR was carried out using Axen^TM^ *Taq* PCR Master Mix (Macrogen, Seoul, Republic of Korea) and gene-specific primers: rabbit SPRN-Forward (5′-GTAAGGCCCAGTGGTGGGAT-3′) and rabbit SPRN-Reverse (5′-GGACTACCGGGATACGGGAT-3′). Primers were designed using the genomic sequence of the rabbit *SPRN* gene that was deposited at GenBank (Gene ID: 100340524). The detailed experimental procedures followed manufacturers’ protocols, using an annealing temperature of 62 °C for 30 s. PCR reactions were purified using a QIAquick Gel Extraction Kit (Qiagen, Valencia, CA, USA) before being used for sequencing analysis. An ABI 3730XL sequencer (Applied Biosystems, Foster City, CA, USA) was used to perform amplicon sequencing of the PCR products. Finch TV Version 1.4.0 was used to read the sequencing data to carry out the genotype analysis.

### 2.4. Statistical Analysis

LD analysis was conducted using the Haploview program (version 4.2, https://www.broadinstitute.org/haploview/haploview) (accessed on 1 February 2024). SAS 9.4 program (SAS Institute Inc., Cary, NC, USA) was used to perform chi-square tests to assess distributional differences in Hardy–Weinberg Equilibrium (HWE), genotype, allele, and haplotype frequencies.

### 2.5. Multiple Sequence Alignments

The information on the Sho protein was acquired from GenBank. This included sequences from humans (*Homo sapiens*, NP_001012526.2), cattle (*Bos taurus*, AAY83885.1), sheep (*Ovis aries*, NP_001156033.1), goats (*Capra hircus*, AGU17009.1), red deer (*Cervus elaphus*, ACF24724.1), dogs (*Canis lupus familiaris*, XP_038296952.1), horses (*Equus caballus*, XP_023492126.1), and rabbits (*Oryctolagus cuniculus*, XP_008268877.2). The amino acid sequences of the Sho protein were aligned using the Geneious Bioinformatics program for Sequence Data Analysis (https://www.geneious.com) (accessed on 1 February 2024) based on a progressive alignment method.

### 2.6. Literature Search

A PubMed literature query was conducted to find reports on *SPRN* polymorphisms in humans, cattle, goats, sheep, horses, and dogs. Query parameters were “prion”, “SNP”, and “polymorphisms” along with “human”, “cattle”, “goat”, “sheep”, “horse”, or “dog”. After initial assessment of the title and abstract, irrelevant investigations were eliminated. The following criteria for participation were assessed by the authors of this manuscript. First, investigations on genetic polymorphisms of *SPRN* were listed, followed by full-text articles. The exclusion standards were (1) case reports and (2) reports with insufficient genotype information.

## 3. Results

### 3.1. Investigation of Polymorphisms in the SPRN Gene of 207 Rabbits

There are two exons in the rabbit *SPRN* gene. The ORF region of the leporine *SPRN* gene was amplified using PCR to investigate genetic polymorphisms. The amplified regions of the *SPRN* gene comprised 1099 bp, including its ORF as well as a small portion of its 3′ UTR. In exon 2 of the ORF region, we discovered two synonymous SNPs, c.129G>T and c.249A>C (Figure 1). The genotype and allele frequencies of these SNPs in the leporine *SPRN* gene are shown in Table 1. The genotype frequencies for the polymorphisms were within HWE. We further examined LD values between the two leporine *SPRN* polymorphisms with |D’| and r^2^ values (Table 2). Notably, strong LD was observed between the leporine *SPRN* polymorphisms, c.129G>T and c.249A>C (|D’|=0.67 and r^2^=1.0). Additionally, we carried out a haplotype analysis for the two polymorphisms of the leporine *SPRN* gene (Table 3). In the leporine *SPRN* gene, the GA haplotype was the most prevalent (0.956), followed by TA (0.05) and TC (0.039) (Table 3). Furthermore, we used |D’| and r^2^ values to examine the LD between *PRNP* and *SPRN* SNPs. Data from a previous study [19] were used to present detailed *PRNP* SNP information. These *PRNP* and *SPRN* SNPs were collected from 207 blood samples of different rabbits for LD analysis, and precise LD values are provided (Table 4). Specifically, *SPRN* SNPs demonstrated exceedingly weak LD with the *PRNP* c.234C>T SNP.

### 3.2. Sequence Alignment of the Sho Protein from Multiple Species

The Sho protein sequence of rabbits was compared with that of TSE-susceptible animals, including humans, cattle, sheep, goats, and red deer, and that of TSE-resistant animals, including dogs and horses. A total of 11 amino acids were unique to rabbits: isoleucine (I) at codon 39, threonine (T) at codon 46, proline (P) at codon 64, proline (P) at codon 88, leucine (L) at codon 107, serine (S) at codon 109, threonine (T) at codon 123, cysteine (C) at codon 130, serine (S) at codon 131, tyrosine (Y) at codon 137, and proline (P) at codon 150 (Figure 2).

### 3.3. Comparing Genetic Polymorphisms of the SPRN Gene among Species

The variations in genetic polymorphisms within the ORF of the *SPRN* gene in rabbits were compared with those of prion disease-susceptible animals (humans, cattle, goats, and sheep) and those of prion disease-resistant animals (horses and dogs). Remarkably, animals susceptible to prion diseases showed more than seven genetic variations in the *SPRN* gene. In contrast, prion disease-resistant animals had only one genetic polymorphism in the ORF region of *SPRN*. Interestingly, our study revealed a unique finding in rabbits: two synonymous SNPs that did not alter the amino acids (Figure 3).

## 4. Discussion

Sho proteins belong to the PrP family and are primarily expressed in the brain, where they have been shown to accelerate the development of prion disorders [20,24]. According to reports, they affect the developmental processes of mammary glands and embryos [25,26]. The genetic features of the *SPRN* gene require further study. In the present study, we examined the genetic features of the *SPRN* gene in rabbits because changes in the genetic makeup of the *SPRN* gene could influence susceptibility to prion disease through structural modifications and protein expression [3,5,21].

In this study, we discovered two novel synonymous SNPs, c.129G>T and c.249A>C, within the ORF of the rabbit *SPRN* gene. Notably, no nonsynonymous SNPs have been found in the ORF of this gene. A recent study has shown that synonymous SNPs can affect mRNA integrity, splicing, and gene transcription [15,27]. Therefore, future investigation into the impact of synonymous SNPs in the rabbit *SPRN* gene on the vulnerability to prion diseases is required. Additionally, the rabbit *SPRN* gene on chromosome 18 displayed strong genetic LDs between the two SNPs, while the *PRNP* and *SPRN* genes in rabbits showed weak LDs. The prion gene family consists of four members, *PRNP, PRND, PRNT*, and *SPRN*, with the first three located on chromosome 13 in cattle [28,29,30,31,32]. Previous studies have demonstrated significant genetic LD among these genes [32,33,34,35].

The *PRNP* SNPs linked to scrapie susceptibility exhibited a strong LD with *PRND* and *PRNT* SNPs. This implies a potential link between SNPs in the *PRND* and *PRNT* genes and susceptibility to prion diseases [17]. However, *SPRN* SNPs showed weak LD values with *PRNP* SNP in this study. Additionally, a previous study found that the dog, a species resistant to prion disease, had low LD values between the *PRNP* and *PRND* gene polymorphisms [27]. Since this effect might be generated by an LD block, the causal relationship between the *PRNP* and *SPRN* genes was not clear until recently. Subsequently, because the *SPRN* and *PRNP* genes are on different chromosomes, prior association investigations for the susceptibility of the *SPRN* gene to scrapie have not been indicative of an LD block affecting the *PRNP* gene [17].

Prior reports have shown that several prion-susceptible species, including cattle, sheep, cats, and goats, exhibit a high degree of nonsynonymous polymorphism in *SPRN* genes, leading to alterations in the amino acid sequence [1,4,5,14,17,22,23,25]. Of note, horses and dogs are prion-resistant species that only have one polymorphism in the *SPRN* gene [6,15,16]. Overall, synonymous sites have been categorized as neutral in terms of functionality. However, recent conflicting research indicates that synonymous alleles may play important roles in a variety of molecular processes. For example, in human disease correlation studies, synonymous SNPs possess an effect size comparable to nonsynonymous SNPs, according to a recent investigation [3]. We found that rabbits lack two amino acids at positions 115 and 116. In addition, there was a deletion/insertion between positions 39 and 43 in ruminants (Figure 3). This could be interesting in terms of the relationship with the folding process of PrP^Sc^ and its connection to susceptibility to prion diseases. Further investigation is highly desirable. In the present study, rabbits were found to have no nonsynonymous SNPs. Although genetic polymorphisms have been examined in a relatively small sample of prion-resistant species compared to prion-susceptible species, the number of prion-resistant species is sufficient to detect rare genetic polymorphisms with afrequency of 1% [6]. The variation in the number of these polymorphisms is evident, which may impact the structure and function of the Sho protein. Consequently, pathogenic changes have a higher probability of developing in a highly polymorphic *SPRN* gene. Further studies are needed to investigate the significance and function of *SPRN* genomic features in prion disease susceptibility across a wider range of species.

Although there is much debate about prion susceptibility and resistance, we evaluated species generally reported to have prion diseases and species that showed resistance to prion diseases using several prion infectivity evaluation methods, including protein misfolding cyclic amplification (PMCA), real-time quacking induced conversion (RT-QuIC), and PrP transgenic mice. However, the definitions of “prion susceptible animals” and “prion resistant animals” are often oversimplified to find prion pathomechanism-related unique genetic factors. In the future, it seems desirable to discover factors related to prion diseases by dividing subjects into groups based on their susceptibility or resistance to prion diseases.

## 5. Conclusions

In conclusion, we identified two synonymous SNPs in the rabbit *SPRN* gene and determined the genotype, allele, and haplotype frequencies of *SPRN* gene polymorphisms in rabbits. Additionally, we observed strong genetic LD between rabbit *SPRN* SNPs located on chromosome 18. Furthermore, our analysis of protein alignment revealed 23 amino acids specific to rabbits. To our knowledge, this is the first investigation of rabbit *SPRN* gene polymorphisms. Since prion disease-susceptible animals have been found to possess multiple non-synonymous SNPs in the SPRN gene, it is apparent that rabbits lack these non-synonymous SNPs.

## Figures and Tables

**Figure 1 animals-14-01807-f001:**
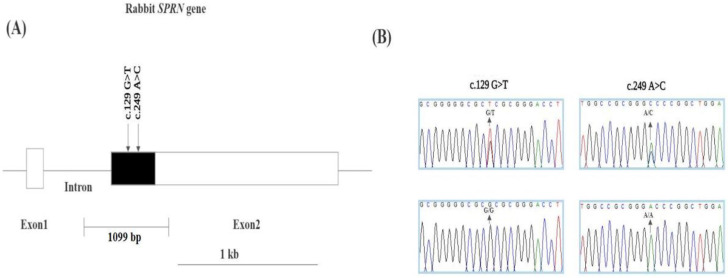
Gene map and electropherograms of single nucleotide polymorphisms (SNPs) of the shadow of prion protein gene (*SPRN*) identified in the rabbit. (**A**) Gene map and novel polymorphisms identified in the *SPRN* gene on chromosome 18 in rabbits. The open reading frame (ORF) within exon 2 is indicated with a black block, and the 5′ and 3′ untranslated regions (UTRs) are indicated with white blocks. Two polymorphisms were found in this study. (**B**) Electropherograms of novel SNPs in the *SPRN* gene. The four colors represent DNA bases in the sequence using an ABI 3730 automatic sequencer (blue: cytosine; red: thymine; black: guanine; and green: adenine).

**Figure 2 animals-14-01807-f002:**
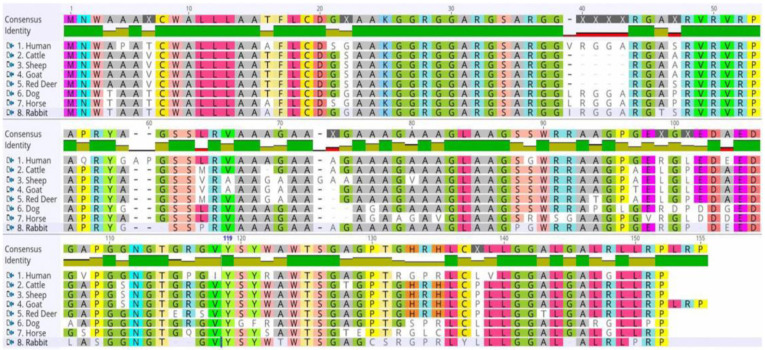
Multiple sequence alignments of the amino acid sequences of various species. Comparison of the amino acid sequence of Sho protein in humans (*Homo sapiens*, NP_001012526.2), cattle (*Bos taurus*, AAY83885.1), sheep (*Ovis aries*, NP_001156033.1), goats (*Capra hircus*, AGU17009.1), red deer (*Cervus elaphus*, ACF24724.1), dogs (*Canis lupus familiaris*, XP_038296952.1), horses (*Equus caballus*, XP_023492126.1, and rabbits (*Oryctolagus cuniculus*, XP_008268877.2). Uncolored regions represent differences among species, whereas colors show the similarity of different amino acids.

**Figure 3 animals-14-01807-f003:**
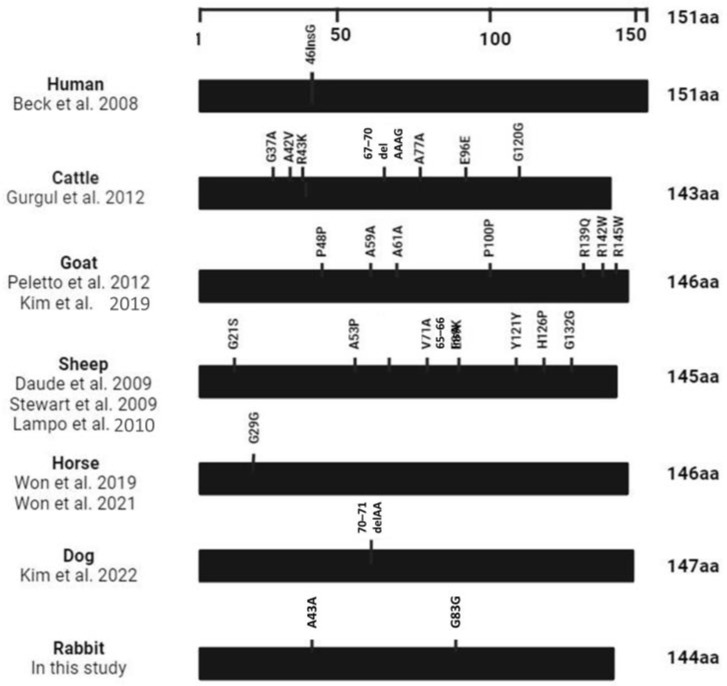
Comparison of genetic polymorphisms in the open reading frame (ORF) of the *SPRN* gene in species susceptible and resistant to prion disease. Polymorphisms of the *SPRN* gene have been reported in previous studies [3,4,5,6,16,17,20,21,22,23]. The length of the amino acids in the *SPRN* gene is depicted by the edged horizontal bar.

**Table 1 animals-14-01807-t001:** Genotype and allele frequencies of *SPRN* gene polymorphisms in rabbits.

SNPs	Genotype	*n*	Genotype Frequency	Allele	*n*	Allele Frequency	PIC	Chi-HW	*P*-HW
G>T	GG	189	0.913	G	396	0.957	0.1058	0.0333	0.9835
	GT	18	0.087	T	18	0.043			
A>C	AA	192	0.929	A	399	0.964	0.0665	2.2214	0.3293
	AC	15	0.071	C	15	0.036			

PIC: polymorphism information content value; Chi-HW: Chi-square value for Hardy-Weinberg equilibrium; *P*-HW: *p* value for Hardy-Weinberg equilibrium.

**Table 2 animals-14-01807-t002:** Haplotype frequency of *SPRN* gene polymorphisms in rabbits.

	c.129 G>T	c.249 A>C	Haplotype Frequency
ht1	G	A	0.956
ht2	T	C	0.039
ht3	T	A	0.005

**Table 3 animals-14-01807-t003:** Linkage disequilibrium (LD) scores between two polymorphisms of the *SPRN* gene in rabbits.

|D’|		
r^2^	c.129 G>T	c.249 A>C
c.129 G>T	-	0.67
c.249 A>C	1.0	-

The figure above the diagonal indicates |D’| value. The figure below the diagonal indicates r^2^ value.

**Table 4 animals-14-01807-t004:** Linkage disequilibrium (LD) scores among single nucleotide polymorphisms (SNPs) of the *SPRN* and *PRNP* genes in rabbits.

|D’|			
r^2^	*PRNP* c.234C>T	*SPRN* c.129 G>T	*SPRN* c.249 A>C
*PRNP* c.234C>T	-	0.0	0.001
*SPRN* c.129 G>T	0.235	-	0.67
*SPRN* c.249 A>C	0.037	1.0	-

The figures above the diagonal indicate |D’| values. The figures below the diagonal indicate r^2^ values.

## Data Availability

The raw data supporting the conclusions of this article will be made available by the authors upon request.

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
