# Peer review of "First Report of Single Nucleotide Polymorphisms (SNPs) of the Leporine Shadow of Prion Protein Gene (SPRN) and Absence of Nonsynonymous SNPs in the Open Reading Frame (ORF) in Rabbits"

_animals, 2024, doi:10.3390/ani14121807_

Round 1

Reviewer 1 Report

Comments and Suggestions for Authors

Manuscript entitled ,,First Report of Single Nucleotide...” represents the first examination of the genetic features of the rabbit SPRN gene. Authors identified rabbit SPRN polymorphisms within a 1099 bp region and calculated LD between the SPRN and PRNP polymorphisms.

- I have a major comment which must be considered. The point is that authors use self-citations too often. For example, the first sentence of Introduction should be supported by review article or original articles concerning these species. As I see, [1]  is original article concerning only cattle. It goes further, the manuscript contains a lot of self-citations, and this should be changed. Personally, I can accept a few, in exceptional situations a little more, but the authors greatly exaggerated. Please also note that self-citations are then presented in each person's scientific statistics, so this is not a good long-term tactic.

- Notice that Abstract should have no more than 200 words.

- New Zealand White and Flemish Giant, not ,,white” and not ,,giant”.

Please improve your manuscript and then I will be able to evaluate it further.

Author Response

Reviewer 1

  1. I have a major comment which must be considered. The point is that authors use self-citations too often. For example, the first sentence of Introduction should be supported by review article or original articles concerning these species. As I see, [1] is original article concerning only cattle. It goes further, the manuscript contains a lot of self-citations, and this should be changed. Personally, I can accept a few, in exceptional situations a little more, but the authors greatly exaggerated. Please also note that self-citations are then presented in each person's scientific statistics, so this is not a good long-term tactic.

Response: Thank for the reviewer’s good comment. As suggested by the reviewer, we have changed the references in the References section [Page 21, lines 2-6; Page 24, lines 3-8].

  1. Notice that Abstract should have no more than 200 words.

Response: Thank for the reviewer’s good comment. As suggested by the reviewer, we have omitted the sentences in the Abstract section [Page 4, lines 7-15 in submitted manuscript].

  1. New Zealand White and Flemish Giant, not ,,white” and not ,,giant”.

Response: Thank for the reviewer’s good comment. As suggested by the reviewer, we have modified information on rabbits in the Materials and methods section [Page 9, lines 8-9].

Reviewer 2 Report

Comments and Suggestions for Authors

My largest concern is the presentation of some animals as being resistant to TSEs and others to be susceptible. I can appreciate that some animals have never been observed to have a TSE. However, the absence of TSE evidence does not indicate evidence of absence. Perhaps there are undiscovered TSEs that can affects rabbits, dogs and horses. For example, camels were not know to be susceptible to TSEs until a new camel prion disease was identified. Perhaps the absence of observed disease reflects limited transmission experiments? In contrast humans are considered susceptible and yet, humans are highly resistant to scrapie. Even BSE, which sadly did transmit to humans, was infrequent, with an estimate 1:50,000 exposed humans contracting the disease.  There are nuances here that are important and sweeping species into resistant and susceptible groups seems too simplistic. 

Was there any association between the sprn SNPs and the breed of rabbits?

Why were these two breeds of rabbits selected? It might have been nice to look at other genera as well.

I may have missed it, but was HWE spelled out the first time that the abbreviation was used?

Line 236 ... chromosome 13 in mice? humans? rabbits?

Line 56-59. I do not think this reference supports that statement. "Crucial in the development of prion diseases" seems too strong. Perhaps soften the statement.

Something strange is happening in the Conclusion section. Lines 272-276.

Author Response

Reviewer 2

  1. My largest concern is the presentation of some animals as being resistant to TSEs and others to be susceptible. I can appreciate that some animals have never been observed to have a TSE. However, the absence of TSE evidence does not indicate evidence of absence. Perhaps there are undiscovered TSEs that can affects rabbits, dogs and horses. For example, camels were not known to be susceptible to TSEs until a new camel prion disease was identified. Perhaps the absence of observed disease reflects limited transmission experiments? In contrast humans are considered susceptible and yet, humans are highly resistant to scrapie. Even BSE, which sadly did transmit to humans, was infrequent, with an estimate 1:50,000 exposed humans contracting the disease. There are nuances here that are important and sweeping species into resistant and susceptible groups seems too simplistic.

Response: Thank for the reviewer’s good comment. As suggested by the reviewer, we have added sentences regarding weakness of the present study in the Discussion section [Page 16, lines 4-11].

  1. Was there any association between the sprn SNPs and the breed of rabbits?

Response: We apologize for the confusion. The rabbits used in this study are one crossbreed rabbits (New Zealand White and Flemish Giant rabbits). We have added the information in the Materials and methods section [Page 9, lines 8-9].

  1. Why were these two breeds of rabbits selected? It might have been nice to look at other genera as well.

Response: We apologize for the confusion. The rabbits used in this study are one crossbreed rabbits (New Zealand White and Flemish Giant rabbits). We have added the information in the Materials and methods section [Page 9, lines 8-9].

  1. I may have missed it, but was HWE spelled out the first time that the abbreviation was used?

Response: Thank for the reviewer’s good comment. As suggested by the reviewer, we have added the full name of HWE in the Materials and methods section [Page 10, line 12].

  1. Line 236 ... chromosome 13 in mice? humans? rabbits?

Response: Thank for the reviewer’s good comment. “… chromosome 13 in cattle” is right. We have added the information in the Discussion section [Page 14, line 18].

  1. Line 56-59. I do not think this reference supports that statement. "Crucial in the development of prion diseases" seems too strong. Perhaps soften the statement.

Response: Thank for the reviewer’s good comment. As suggested by the reviewer, we have modified the sentence in the Introduction section [Page 6, line 11].

  1. Something strange is happening in the Conclusion section. Lines 272-276.

Response: Thank for the reviewer’s good comment. We have modified the sentences and format in the Discussion section [Page 16, lines 6-9].

Reviewer 3 Report

Comments and Suggestions for Authors

Thank you for allowing me the opportunity to read your manuscript, comments to the Author

- why only coding region was amplified? Why authors did not included in sequencing UTRs as they sequenced intron ?

- Authors stated that they used two phenotypically differnt rabbits breeds:

a) how many New Zealand White and Flemish Giant were used in study?

b) If You used two breeds in your experiment why it is not stated in tables? There is visible low representation of allel T in first SNP and allel C in second SNP - they were identified in both population or only in one of them?

part 4.2 - based on which algorithm or tool you stated that 11 amino acids are unique? what about positions 115 and 116 - in rabbits there is lack of two aminoacids. positions between 39 and 43 - it seems that in ruminants there is deletion/insertion - maybe this could intrestig in case of folding of aminoacids sequence and have something with susceptibility for prior disease? could you comment this? especially that first SNP was found on aminoacids position 43 if i undrstand correctly

figure 3 - description of rabbit representation of aa sequence is unclear - what is 43AA and G83G?

lines 270 -273 - correct style

why in literature positions 9, 10 and 13 are underlined?

Author Response

Reviewer 3

  1. Why only coding region was amplified? Why authors did not include in sequencing UTRs as they sequenced intron?

Response: Thank you for the reviewer’s good comment. Because the SPRN gene has a high GC content, we tried several amplification conditions and found that it was difficult to amplify the ORF and 3' UTR together. Therefore, this study was focused on determining the optimal conditions for amplifying ORFs.

  1. Authors stated that they used two phenotypically different rabbits breeds: how many New Zealand White and Flemish Giant were used in study?

Response: We apologize for the confusion. The rabbits used in this study are one crossbreed rabbits (New Zealand White and Flemish Giant rabbits). We have added this information in the Materials and methods section [Page 9, lines 8-9].

  1. If you used two breeds in your experiment why it is not stated in tables? There is visible low representation of allele T in first SNP and allele C in second SNP - they were identified in both population or only in one of them?

Response: We apologize for the confusion. The rabbits used in this study are one crossbreed rabbits (New Zealand White and Flemish Giant rabbits). We have added the information in the Materials and methods section [Page 9, lines 8-9].

  1. part 4.2 - based on which algorithm or tool you stated that 11 amino acids are unique?

Response: Thank you for the reviewer’s good comment. The amino acid sequences of the Sho protein were aligned using the Geneious Bioinformatics program based on a progressive alignment method.

  1. What about positions 115 and 116 - in rabbits there is lack of two amino acids. positions between 39 and 43 - it seems that in ruminants there is deletion/insertion - maybe this could be interesting in case of folding of amino acids sequence and have something with susceptibility for prion disease? could you comment this? especially that first SNP was found on amino acids position 43 if i understand correctly

Response: Thank you for the reviewer’s good comment. We completely agree that the content presented by the reviewer is very interesting. We have added the sentences in the Discussion section [Page 15, lines 14-18].

  1. Figure 3 - description of rabbit representation of aa sequence is unclear - what is 43AA and G83G?

Response: We apologize for the confusion. ‘43AA’ indicates ‘A43A’. A43A and G83G are two synonymous SNPs found in this study. We have changed ‘43AA’ to ‘A43A’ in Figure 3.

  1. lines 270 -273 - correct style

Response: Thank you for the reviewer’s good comment. We have modified the styles in the Conclusion section [Page 16, lines 6-9].

  1. Why in literature positions 9, 10 and 13 are underlined?

Response: Thank you for the reviewer’s good comment. We have omitted the underlines.

Round 2

Reviewer 1 Report

Comments and Suggestions for Authors

In my opinion, manuscript is now corrected, especially that authors also commented on the comments of other reviewers.

Well done!

Author Response

1. In my opinion, manuscript is now corrected, especially that authors also commented on the comments of other reviewers. Well done!

Response: Thank you for the reviewer’s comment.